# The Contribution of Visual and Auditory Working Memory and Non-Verbal IQ to Motor Multisensory Processing in Elementary School Children

**DOI:** 10.3390/brainsci13020270

**Published:** 2023-02-05

**Authors:** Areej A. Alhamdan, Melanie J. Murphy, Hayley E. Pickering, Sheila G. Crewther

**Affiliations:** 1Department of Psychology, Counselling and Therapy, La Trobe University, Melbourne, VIC 3086, Australia; 2Department of Psychology, Imam Mohammad Ibn Saud Islamic University, Riyadh 11564, Saudi Arabia; 3Centre for Human Psychopharmacology, Faculty of Health, Arts and Design, School of Health Sciences, Swinburne University of Technology, Hawthorn, VIC 3122, Australia

**Keywords:** children, auditory, visual, audiovisual, multisensory processing, motor reaction times, working memory, non-verbal intelligence

## Abstract

Although cognitive abilities have been shown to facilitate multisensory processing in adults, the development of cognitive abilities such as working memory and intelligence, and their relationship to multisensory motor reaction times (MRTs), has not been well investigated in children. Thus, the aim of the current study was to explore the contribution of age-related cognitive abilities in elementary school-age children (*n* = 75) aged 5–10 years, to multisensory MRTs in response to auditory, visual, and audiovisual stimuli, and a visuomotor eye–hand co-ordination processing task. Cognitive performance was measured on classical working memory tasks such as forward and backward visual and auditory digit spans, and the Raven’s Coloured Progressive Matrices (RCPM test of nonverbal intelligence). Bayesian Analysis revealed decisive evidence for age-group differences across grades on visual digit span tasks and RCPM scores but not on auditory digit span tasks. The results also showed decisive evidence for the relationship between performance on more complex visually based tasks, such as difficult items of the RCPM and visual digit span, and multisensory MRT tasks. Bayesian regression analysis demonstrated that visual WM digit span tasks together with nonverbal IQ were the strongest unique predictors of multisensory processing. This suggests that the capacity of visual memory rather than auditory processing abilities becomes the most important cognitive predictor of multisensory MRTs, and potentially contributes to the expected age-related increase in cognitive abilities and multisensory motor processing.

## 1. Introduction

Numerous psychophysical and neuroimaging studies have established a consistent association between cognitive abilities such as working memory (WM) [1], intelligence, and motor development in both adults [2] and children [3,4,5]. Indeed, as early as 1988, Haier et al. demonstrated using positron emission tomography that smart adult brains process visually based information faster and require fewer nutritional resources [6]. More recent fMRI studies have also provided evidence that cognitive and motor development are interrelated and are mediated by the concurrently co-activated dorsolateral prefrontal cortex and the neocerebellum, during both cognitive and motor tasks [7,8,9]. Research has also supported age-related improvements in multisensory motor reaction time (MRT) tasks [10] and cognitive functions, showing an accelerated developmental progression during later childhood [3,5]. Indeed, cognitive abilities such as WM [11], and fluid or general intelligence (IQ) [12,13] are thought to be associated with the increase in multisensory MRTs seen with age, yet the link between higher cognitive functioning and multisensory MRTs in children has seldom been investigated. Although it is well accepted that in adults WM, short-term memory (STM), and IQ are associated [14,15,16], studies on the relationship between STM, WM, and intelligence in children have not distinguished between verbal and spatial STM and WM [15], nor the relationship of such skills to MRTs in early school years. Thus, the primary aim of this study was to investigate the development of visual and auditory STM and WM performance congruently with MRT processing in young school-aged children, employing commonly used experimental measures of multisensory motor abilities that are known to increase across childhood, including the audiovisual multisensory detection task and visuomotor processing tasks (i.e., [10,17,18,19]).

WM has traditionally been defined as the memory system responsible for actively maintaining current information for a short period of time, allowing for it to be manipulated and accessed either in the present moment or later, and is suggested to support and underlie many complex processes such as learning, reasoning, and problem solving [20,21,22]. STM is also considered an interactive component of WM, which refers to a capacity-limited memory system involved in the brief storage of information received from either verbal or visuospatial representations [23]. WM is also often considered to be a component of nonverbal intelligence [24], or perhaps even synonymous [25]. Conway and Kovacs also found that tests of non-verbal fluid intelligence, such as Raven’s Progressive Matrices, were positively correlated with estimates of WM capacity on tasks that required the simultaneous storage and processing of information [26]. The association between performance on Raven’s Progressive Matrices and WM capacity is likely to vary with the level of item difficulty [16], as according to Raven’s manual [27] the easiest items are presented early in the test, and the hardest items are presented last, with fewer than 10% of appropriately aged participants likely to be able to solve the hardest items [16]. Indeed, Corman and Budoff classified the RCPM into four factors based on item difficulty, with the easiest factor being “simple continuous pattern completion”, and the “reasoning by analogy” factor being the most sophisticated level of cognitive processing [28] and likely to require the activation of WM [16,29]. 

In addition, a study conducted by Schear and Sato investigated an information-processing model which hypothesized that information-processing domains requiring the integration of vision and motor speed would significantly contribute to performance on complex cognitive tests such as the digit symbol subtest [30]. The authors also noted that time to complete the motor component of the visuomotor pegboard task but not visual acuity alone strongly contributed to the complex cognitive tests requiring vision, motor speed, and dexterity [30]. It has also been shown that children with faster RT in multisensory processing achieve higher intelligence scores [3,12,31], and show better WM capacity [11,32]. However, none of the above behavioural studies have systematically investigated the relationships among visual and auditory STM and WM, nonverbal intelligence, and multisensory motor performance.

Although the literature appears to support the association between the development of multisensory motor processing and WM, research has mainly focused on auditory verbal short-term storage and/or WM manipulation, and not from the viewpoint of visual WM performance. For example, Denervaud et al. used audiovisual motor detection tasks to demonstrate the relationship between multisensory gain in a simple detection task and cognitive measures such as auditory WM and intelligence [32]. The authors found that children’s MRTs predicted auditory digit span WM scores (*p* = 0.02) and fluid intelligence (*p* = 0.03). However, Barutchu and colleagues reported no correlation between multisensory reaction time and auditory WM scores associated with auditory digit span backwards [3,12]. Due to this inconsistency of previous studies, as well as the exclusive focus onauditoryWM, it is important to further investigate how the specific sensory domains of WM (i.e., visual and auditory aspects) might contribute to age-related differences in multisensory motor processing. 

Indeed, there is evidence that visual WM develops significantly during infancy and early childhood, and that the developmental changes in both visual STM and WM are associated with major gains in visual attention, perception, and language [33,34]. A study that employed an 18-month follow-up involving primary school-aged children found that children with faster visual WM (assessed by a one-back task) had better fine motor skills [5]. In addition, the associations between visuospatial and auditory WM and visuomotor reaction time have also been found in adult literature [2] with these authors finding that visual WM—but not auditory- WM—explained a significant portion of the variance in the rate of visuomotor reaction time performance. 

To date, there is little agreement on how visual and auditory WM together with nonverbal IQ contribute to multisensory MRT measures, particularly in young school-aged children. Thus, the current study aimed to use Bayesian analyses to examine the concurrent cognitive performance of visual and auditory STM and WM (auditory and visual digit span), and nonverbal IQ as assessed by Raven’s Coloured Progressive Matrices (RCPM), on MRT measures of multisensory (auditory, visual, and audiovisual), and visuomotor processing across different educational profiles (Prep, Grade 1, Grade 2, and Grades 3 and 4). The specific aims were:(i)to investigate the apparent concurrent developmental changes in classical measures of WM, such as visual and auditory digit span, and nonverbal IQ (RCPM);(ii)to investigate developmental changes in the associations between age, nonverbal IQ (RCPM), visual and auditory STM and WM, and multisensory processing when measured by MRTs; and(iii)to determine how visual and auditory STM and WM and nonverbal IQ contribute to MRTs for multisensory processing.

It was hypothesized, in line with past research [35,36], that older children would demonstrate both longer forward and backward digit spans, and faster MRTs on multisensory tasks [10,17]. It was also hypothesized that combined cognitive development as a measure of WM and IQ tasks would contribute further to MRTs of multisensory processing. Additionally, we expected to see strong correlations between MRT tasks and uni- and multisensory information detection tasks and cognitive tasks such as WM and nonverbal IQ.

## 2. Method

### 2.1. Participants

The study included a total of 75 participants from the preparatory/foundation year to Grade 4: Prep (*n* = 18), Grade 1 (*n* = 11), Grade 2 (*n* = 20) and Grade 3+4 (*n* = 26) (see Table 1). The children were recruited from Catholic and public elementary schools in Victoria, Australia. The Victorian Department of Education approved the project, and the individual school principals assisted in distributing information and consent forms to the parents and guardians of the children. This study was approved by the La Trobe University Human Ethics Committee (HEC 18139, HEC 16121), the Victorian Department of Education Human Ethics Committee, and the Victorian Catholic Schools Ethics Committee. All children of parents/guardians who had signed the forms indicating consent for their child to participate in the project, and who had completed a brief questionnaire on medical health and neurodevelopmental anomalies, were included in testing for the study. However, only children aged 5–10 with normal or correct-to-normal vision and hearing, with no history of clinically diagnosed neurodevelopmental disorders such as ADHD, Autism spectrum disorder, language disorder, or intellectual disability were included in the analyses of the study. According to the Declaration of Helsinki, parents and children were entitled to withdraw a child’s participation or data at any time. Verbal assent was also obtained from each child prior to each testing session. A flowchart of the eligibility criteria, participant groups, and experimental series is shown in Figure 1.

### 2.2. Screening and Psychometric Measures

#### 2.2.1. Vision and Hearing Screening 

Screening for vision and audition was conducted to determine whether children had normal hearing and normal or corrected-to-normal vision. During the vision screening, Snellen charts were used to assess distance and near visual acuity, whereas the Ishihara test was used to assess colour vision. Screening for auditory ability through each ear was carried out on a commercial audiometer (Interacoustic Screening Audiometer, portable audiometer model AS208) in accordance with the Guideline for Hearing Screening in the School Setting, Missouri Department of Health and Senior Services Division of Community and Public Health using Peltor H7A sound attenuating headphones. Sound frequencies ranging from 250 Hz to 8000 Hz and sound pressure levels (SPL) at each octave were assessed. During testing, the children were instructed to raise their hand on the same side as the sound and place it down when the sound ceased. 

#### 2.2.2. Nonverbal Intelligence (RCPM) 

Raven’s Coloured Progressive Matrices test (RCPM) was used to assess non-verbal intelligence [37]. The RCPM is a relatively quick, well-normed, highly reliable (test-retest *r* = 0.80) [38], and culture-free psychometric test of nonverbal reasoning abilities in children aged 5–11 years [38,39]. In this test, performance requires cognitive manipulation based on visual icons rather than auditory or lexical choices. The RCPM consists of 36 coloured matrices divided into three sets (A, Ab, B), each comprising 12 problems increasing in complexity and difficulty. The participant is asked to complete the matrix by identifying the most appropriate solution out of six alternative options. Four distinct factors of intellectual abilities are measured by the RCPM Factor 1: Completion of Simple Continuous Patterns, Factor 2: Completion of Discrete Patterns, Factor 3: Continuity and Reconstruction of Simple and Complex Structures, and Factor 4: Reasoning by Analogy [28,40].

### 2.3. Experimental Measures

#### 2.3.1. Multisensory Task

Multisensory processing was measured using motor reaction times to target detection. The targets included three types of stimuli: an auditory stimulus (AS; beep), a visual stimulus (VS; grey circle), and an audiovisual stimulus (AVS; beep and grey circle presented simultaneously) (see Figure 2). The procedure selected for use was similar to the one used by [17] and our recent study [10]. The stimuli were presented and controlled using VPixx^TM^ software (V 3.20) and RESPONSEPixx (VPixx, Vision Science Solutions, Quebec, Canada). The children were instructed to press a button from the button box on the handheld RESPONSEPixx box (developed by Peter April (http://www.vpixx.com/, accessed on 1 June 2019)) as rapidly and accurately as possible to indicate the stimulus and record their responses. Prior to testing, practice trials were conducted for each condition (AS, VS, and AVS) to ensure that all children could understand the procedure and performed accurately and quickly, especially the youngest first-year group. Auditory stimuli consisting of a 1500 Hz tone with a rise and fall time of 5 ms were presented through closed headphones. Visual stimuli were presented as a Gaussian circle with variable peripheral target locations (i.e., never positioned centrally), to ensure the maintenance of conscious attention to completion. The mean motor reaction times (i.e., the time taken between the onset of the stimulus and button press) were extracted from each condition of the multisensory task. An interstimulus interval of 1500–2500 ms with a duration of 150 ms was applied to all the trials. In terms of internal reliability, Cronbach’s alpha for the AS, VA, and AVS reached a total of 0.93, indicating high reliability [10].

#### 2.3.2. Visuomotor Processing using the SLURP Eye-Hand Coordination App

To assess fine visually driven motor (visuomotor) processing, the Lee-Ryan Eye-Hand Coordination Test battery (SLURP) was used [41]. It has been demonstrated that this task is reliable and valid for assessing visuomotor integration in both children and adults [18,35]. In this task, children were instructed to trace five shapes in order (circle, triangle, square, rabbit, and snail); the total time taken to accomplish the task was extracted and analysed for each child. In order to demonstrate how the test would be conducted, and to ensure no order effect across the test items, participants first completed the “Castle” item (see Figure 3). This item was chosen as a practice as it requires many changes in direction over a considerable distance [18].

#### 2.3.3. Visual and Auditory Digit Span (Forward and Backward)

The forward and backward digit span tasks were adapted from the auditory digit span subtest of the Wechsler Intelligence Scale for Children—Fifth Edition (WISC-V). The forward digit span task is considered as the measure of immediate recall of sensorily presented information from short-term memory, whereas the digit span backward task requires the manipulation of that information (i.e., reordering) which is usually considered to measure WM abilities [42]. This task was administered in two modalities, visually and auditorily. In the visual digit span condition, digits were presented on a computer screen (black Ariel 92pt font on a white background) [43,44,45]. In the auditory digit span condition, the researchers presented digits orally at a rate of one digit per second without a visual representation of the numbers, and the children were instructed to repeat the digits orally. The children were then asked to repeat the digits either in the same order (forwards condition; STM) or in reverse order (backward condition; WM). The forward condition always preceded the backward condition. The task always began with two trials with a sequence length of two digits. Sequences were progressively increased until a child answered two trials of the same length incorrectly. The children’s longest span (longest sequence answered correctly) is their score, indexing maximum short-term and working memory capacity.

### 2.4. Procedure

The children were assessed individually, typically over four sessions limited to 20–30 min sessions to ensure task engagement and minimise fatigue, in the presence of at least two researchers during school hours in a quiet private room. Testing was initiated by vision and hearing screening followed by experimental tasks preceded by adequate practice trials. The data of two children (one in the Prep year, and one in the Grade 1 group) whose error score was greater than 50% in either the AS or VS trials were excluded (see Figure 1 for details). As this study involved young children in their first year of formal school, the researchers encouraged the children to take frequent breaks. As a thank you gift for their participation, a sticker or small item of stationery was given to each child at the end of each session. 

### 2.5. Data Analysis

The sample size was determined via power analysis using the G*Power 3.1 analysis software [46]. This indicated that a total sample size of 32 participants was required for one-way ANOVAs to achieve a moderate effect size at α < 0.05 at a power of 0.8 (1-β error probability) as suggested by [47]. We achieved this power and exceeded it in each ANOVA, obtaining a power of 0.9 (1-β error probability).

A Bayesian statistical approach was used for all data analyses using the free software JASP 0.16.3.0 ([48]; http://www.jasp-stats.org/, accessed on 1 July 2022). We chose to use a Bayesian approach as it relies on a model comparison rationale and employs a model selection strategy to quantify the strength of evidence for and against each model [49,50] rather than null hypothesis testing models underpinning frequentist statistics. Furthermore, Bayesian statistics have been reported to allow multiple statistical tests to be conducted without increasing the risk of first-type errors [51]. Higher Bayes factors (BF_10_) are interpreted as evidence in favour of the alternative hypothesis compared to null hypothesis testing. The interpretation of BF_10_ values was in accordance with Wetzels and Wagenmakers as anecdotal evidence (1–3), moderate evidence (3–10), strong evidence (10–30), very strong evidence (30–100), and extreme or/decisive evidence if >100 [52]. We acknowledge that it is a general convention of frequentist statistics to report significance to two decimals when reporting statistical outcomes. However, unlike *p* values in frequentist statistics, the Bayes factor in Bayesian statistics provides an indication of the strength of the evidence (effect size) for the alternative (experimental) hypothesis against rival (prior) models meaning that there is value in reporting to the third decimal, which is in line with the Bayesian Analysis Reporting Guidelines and JASP Bayesian reporting guidelines [53,54] to ensure transparency and allow for the critical appraisal of our hypothesis. 

The data were analysed using Bayesian ANOVA, correlation, and multiple linear regression. First, a series of Bayesian one-way ANOVAs were performed to determine whether there was evidence for differences in performance between the grades on the non-verbal IQ (RCPM), visual and auditory STM, and visual and auditory WM. Post hoc comparisons were calculated for each Bayesian ANOVA using a default t-test with a Cauchy prior [55]. The prior and posterior odds and 95% credible intervals (95% CI) are reported. Omega-squared (
$\omega^{2})$
 was also calculated for the ANOVAs in order to estimate the effect size (ES) for the differences between the grades and has been suggested to avoid biased estimations of variance across the design [56,57]. Effect sizes were reported as: 
$\omega^{2}$
 > 0.01 = small; 
$\omega^{2}$
 > 0.06 = moderate; 
$\omega^{2}$
 > 0.14 = large [58]. Second, Bayesian correlations were conducted to explore the relationships between the nonverbal IQ and visual and auditory STM and WM tasks using a default prior (stretched beta prior width = 1) to compute the Bayes factors (BF). The Pearson correlation coefficient (r), the Bayes Factor (BF_10_), and credible intervals (95% CI) are reported. Lastly, Bayesian linear regression analyses were performed to determine which model (combination of predictor variables) indicated the largest degree of predictive evidence for auditory MRTs, visual MRTs, audiovisual MRTs, and the visuomotor task, with the best-fitting model being that with the highest Bayes Factor (BF). In each regression analysis, we entered the non-verbal IQ (RCPM), visual and auditory STM and WM tasks as predictor variables. For each regression model, we reported the “P(M) column” = prior model probability, “P (M|data) = the updated probabilities after having observed the data for each model, “BF_M_” = improvement in the model after seeing the data, “BF” = the Bayes factor compared to the best fitting model (i.e., a value of 1 indicates the best model), and “R^2^” = the percentage of variance We also reported “95% credible intervals (95% CI)” and “BF _inclusion_”, which suggested that values higher than 1 showed evidence to be included as predictors (see [59] for more details).

## 3. Results

### 3.1. Results 1: Differences in Visual and Auditory Short-Term and Working Memory Tasks and Nonverbal IQ across Grades

To determine whether there were grade differences based on the educational profile (Prep, Grade 1, Grade 2, and Grades 3 and 4) in visual and auditory STM and WM (digit span forward and backward) tasks and nonverbal IQ, a series of Bayesian one-way ANOVAs were performed. Descriptive statistics for all dependent measures are shown in Table 2.

For the Visual Digit Span Forward (VDSF) and Backward (VDSB) tasks, the results of VDSF showed very strong evidence for differences across grades in favour of the alternative hypothesis (BF_10_ = 51.216, 
$\omega^{2}$
 = 0.19), indicating there were significant differences between the grades. Post hoc analysis showed anecdotal to moderate evidence for greater performance for Grades 3 and 4 compared to Prep and Grade 1, whereas no difference was observed between Prep, Grade 1, and Grade 2 (see Figure 4a, Table 3a). For *VDSB*, our results also demonstrated decisive evidence of the alternative hypothesis (BF_10_ = 86,082.561, ω^2^ = 0.37), and again these anecdotal to decisive differences were driven by children from Grades 3 and 4 and Grade 2 performing better than those in Prep and Grade 1 (see Figure 4b, Table 3b). 

For the Auditory Digit Span Forward (ADSF) and Backward (ADSB) tasks, the results indicated strong differences across the grades, thus supporting the alternative hypothesis that shows significant differences in performance between grades (BF_10_ = 20.599, ω^2^ = 0.16, BF_10_ = 22.472, ω^2^ = 0.17) for *ADSF* and *ADSB*, respectively. Post hoc analysis, however, showed no evidence to anecdotal differences between grades for both auditory digits forward and backward (see Figure 4c,d, Table 3c,d).

Bayesian one-way ANOVA for nonverbal IQ (RCPM) revealed significant differences (i.e., decisive evidence) across grades that supported the alternative hypothesis (BF_10_ = 1.144 × 10^10^, ω^2^ = 0.54). Post hoc comparisons showed that these differences were driven by the children from Grades 3 and 4 performing decisively better than the children from Prep and Grade 1. The children in Grade 2 also performed decisively better than those in Prep. However, there was only anecdotal evidence of differences between the children in Prep, Grade 2, and Grade 1 (see Figure 4e, Table 3e). Additional analyses of differences based on grades for nonverbal IQ (RCPM) were dependent on Raven’s item difficulty as associated with Corman and Budoff’s factors [28] (see Appendix A).

### 3.2. Results 2: Relationships among Age, Nonverbal IQ, MRTs and Visual and Auditory Short-Term and Working Memory Tasks

Bayesian correlations were performed across the total sample to investigate the evidence of associations using the Bayes Factor (BF) between age, nonverbal IQ, multisensory MRT tasks, and STM and WM tasks. The results revealed evidence for correlations between chronological age and all dependent measures in favour of the alternative hypothesis, with the more complex visually based tasks such as RCPM and VDSB tasks showing a more decisive and significant correlation (*r* = 0.74–0.80) with age. In addition, there was also very strong to decisive evidence between higher performance on nonverbal IQ and faster MRT tasks as well as a greater capacity for visual and auditory forward and backward digit span. The results also showed that VDSB was very strongly to decisively correlated with all MRT tasks, suggesting that better performance on the VDSB task is associated with faster multisensory MRTs. Further, only anecdotal evidence of relationships between auditory ADSF and ADSB tasks and multisensory MRT measures was found (Table 4). When we classified Raven’s into four factors according to Corman and Budoff [28], the results revealed strong to decisive evidence between factor 3 (continuity and reconstruction) and factor 4 (reasoning by analogy) and multisensory MRT tasks and visual and auditory forward and backward digit span (Table 5).

Bayesian correlational analyses were also performed on each grade separately to understand the associations between our measures at each grade level. The results revealed that there was no evidence of associations between nonverbal IQ, visual and auditory forward and backward, and multisensory MRT tasks for Prep, Grade 1, and Grade 2. For children in Grades 3 and 4, there was anecdotal to moderate evidence of the association between nonverbal IQ, WM measures, and multisensory MRT tasks, which supports the alternative hypothesis, suggesting that better performance on visual and auditory WM tasks is more likely to be associated with faster multisensory MRTs of AS, VS, and AVS in the older children. Full correlation tables for each grade for all dependent measures are available in Appendix A.

### 3.3. Results 3: Contribution of Visual and Auditory Working Memory and Nonverbal IQ to MRTs to Auditory, Visual and Audiovisual, and Visuo-Motor Stimuli

We performed additional Bayesian linear regression to investigate the extent to which working memory tasks (visual and auditory short-term and working memory) predict motor multisensory processing. Table 6 presents the results of four regression models investigating non-verbal IQ (RCPM), visual short-term digit span (VDSF), visual working memory (VDSB), auditory short-term digit span (ADSF), and auditory working memory (ADSB) scores as predictors of auditory RT, visual RT, audiovisual RT, and total time to complete each item on the visuomotor task.

In the first regression, using performance scores on non-verbal IQ and short-term and working memory tasks to predict MRTs for auditory stimuli indicated that among all possible models, the best predictive model was for both visual STM and WM (VDSF+ VDSB). After observing the data, the odds in favour of the model containing both VDSF and VDSB as a predictor increased by a factor of 4.952, and this model was 1.34 times more likely than the model with the next-highest BF_10_ value. Further inspection of the posterior inclusion Bayes factor (BF_inclusion_) showed anecdotal to moderate evidence supporting the inclusion of VDSF and VDSB as predictors of auditory MRTs. 

In the second regression, MRTs for visual stimuli were regressed on the same predictors as in the first model. After observing the data, the model containing nonverbal IQ and VDSB was the best model, showing the odds in favour of the model containing IQ and VDSB as a predictor to have increased by a factor of 13.01. This model was 2.61 times more likely than the model with the next-highest BF_10_ value. The posterior summary suggested strong evidence for nonverbal IQ and anecdotal evidence for the VDSB for inclusion in this model as predictors. 

Similarly, in the third regression, the MRTs for audiovisualwere regressed on the same variables, with the model of nonverbal IQ + VDSB also supported as the best model. Similarly, after observing the data, the odds in favour of the model containing IQ and VDSB as a predictor increased by a factor of 10.91, with this model 1.83 times more likely than the model with the next-highest BF_10_ value. The posterior summary suggested that there is evidence for the inclusion of nonverbal IQ (very strong) and VDSB (anecdotal to moderate) as predictors. Thus, nonverbal IQ together with VWM (VDSB) predictors made a unique contribution to multisensory MRTs. 

In the last regression, the visuomotor (SLURP) was also regressed on non-verbal IQ, STM and WM task performance. The model containing visual STM and WM (VDSF+VDSB) was supported as the best model. Observing the data showed that this order increased the odds in favour of the model by a factor of 5.13, making this model 1.07 times more likely than the next model including VDSB and nonverbal IQ. The posterior summary of both models suggested anecdotal to moderate evidence for the inclusion of VDSF, VDSB, and nonverbal IQ as predictors. Table 6 shows the best models for each regression analysis, and Table 7 displays a summary of the regression coefficients of the five multiple regression analyses.

To summarize, our results comparing grade differences show that performances on visually-based tasks such as visual digit span forward (VDSF) and backward (VDSB, and nonverbal IQ (RCPM), are significantly different, supporting the alternative hypothesis. However, auditory digit span forward (ADSF) and backward (ADSB) tasks showed no evidence of grade differences. In addition, Bayesian correlation showed decisive evidence of age-related correlations between the RCPM scores and item difficulty, visual WM, and multisensory MRT measures, but no significant correlations with auditory WM and multisensory MRTs. Finally, Bayesian regression demonstrated that visual STM and WM together with nonverbal IQ consistently predicted multisensory MRTs for AS, VS, AVS, and time to complete the SLURP visuomotor processing task.

## 4. Discussion

The aims of this study were to investigate developmental changes in cognitive measures of visual and auditory STM and WM and nonverbal IQ, and to investigate the predictive contribution of these skills to multisensory MRTs in school-aged children. The main findings from our Bayesian analyses revealed significant and very strong to decisive evidence for grade differences in visual STM and WM capacity, whereas auditory STM and WM capacity showed no significant differences across the grades. In addition, nonverbal IQ performance showed decisive evidence for significant age-related improvement in correct scores on later items of the RCPM as the test increased in difficulty (full results incorporating Raven’s item difficulty factors can be found in the Appendix A). Furthermore, we found decisive evidence of age-related correlations between visual WM and multisensory MRT measures but no significant correlations between auditory WM and multisensory MRTs. Finally, visual STM and visual WM together with nonverbal IQ consistently predicted multisensory MRTs for AS, VS, AVS, and time to complete the SLURP visuomotor processing task. Such results suggest that vision plays a key role in age-related increases in cognitive abilities and in the speed of multisensory motor processing. The results will be discussed first according to the grade differences in the measures of visual and auditory WM and nonverbal IQ, followed by a discussion of the relationships and the contribution of WM and nonverbal IQ to multisensory MRTs.

### 4.1. Age Group Differences in Visual and Auditory Memory and Nonverbal IQ

Consistent with our hypotheses, there was very strong to decisive evidence for grade differences in visual STM and WM (forward and backward digit span), but not in auditory STM and WM (forward and backward digit span). Previous research has also reported significant grade/age-group differences for both visual and auditory WM in school-age children [35,61,62,63,64], which is not fully reflected in the results of the current study, where we found decisive grade differences in visual WM but not auditory WM. In line with our findings, Buss et al. also reported that visual WM improves rapidly across infancy and early childhood [33], with much of the age-related improvement in visual attention and rapid perceptual processing [65] associated with the efficacy of eye movements [33,66] and rapid anatomical brain growth during this period [67]. There is also evidence from adult neuroimaging studies demonstrating that the visual WM system involves neural networks and areas associated with the visually driven goal-directed parieto-frontal network [68] and temporal cortex. 

Furthermore, more recent cognitive neuroscience models for visual perception indicate that, although there are multiple interconnections between the two major functional visual streams (i.e., dorsal and ventral streams [69,70]), different visuomotor subpathways also exist within the longitudinal fasciculi of the dorsal stream [71], with the ventrodorsal stream playing a role in the online control of action, and the dorso-dorsal stream being involved in higher-level cognitive processes such as action understanding [69,72]. Importantly, a recent review has reported that the neural maturation of the visual system results in an improvement in a variety of visual skills such as visual exploration, visual field awareness, and motion sensitivity, as well as cognitive abilities such as attention, working memory, and visuomotor eye–hand coordination [69]. Furthermore, age-related increases in activity in the frontal areas during visual WM tasks and increasing task demand have been demonstrated in children [33,67,73]. In addition, knowledge and experience may also provide further explanation for the decisive differences in performance on visual WM tasks in our study. More specifically, as children become older, their knowledge (i.e., processing strategies) [74] and experience (i.e., familiarity with the task) [75] are considered to contribute to the ongoing development of visual WM.

Nonverbal IQ, as measured with the RCPM, also showed significant (i.e., decisive) differences between the grades, indicating that visually assessed nonverbal IQ develops significantly during early childhood, with children showing progressive development of a mature problem-solving approach, while in turn showing improved processes of complex pattern matching and visual reasoning [38,40]. When we categorised the items of the RCPM into four factors, Bayesian evidence highlighted grade differences in the more complex items such as those in Factor 3: Continuity and Reconstruction of Simple and Complex Structures. This is consistent with our hypothesis and earlier lab research [40] showing that children (6–11 years) made more errors on the hardest items due to an increase in task difficulty [38,76].

### 4.2. Relationships among Age, Nonverbal IQ, Visual and Auditory Working Memory, and MRT Multisensory Measures

Overall, our findings demonstrated *decisive evidence* for the associations between visually assessed nonverbal IQ (RCPM), MRTs for VS and AVS, and visual WM tasks across grade levels. This is in line with past research indicating that higher-order visually driven cognitive processes such as sustained attention, WM, and vocabulary develop and mature during childhood [34,38], which is similar to multisensory motor processing that also continues to improve until late adolescence [10,77]. These results are also in line with the information processing model [30] which suggests that cognitive abilities involving problem solving and the manipulation of information together with sensory motor speed and dexterity factors are related to vision. Indeed, some theories propose that intelligence and WM are primarily driven by general information processing speed in adults (e.g., [78,79,80]). According to such views, people with high intelligence scores and better WM capacities would generally be faster on simple and choice reaction time tasks [6].

Indeed, in adults, there is evidence that the improvements on multisensory MRT and serial reaction time (SRT) tasks are linked to the visual WM capacity [2,81,82]. However, there is relatively little child research examining the specific associations between either visual or auditory WM and multisensory MRTs, with Barutchu and colleagues [3,12] also failing to find a systematic correlation between measures of multisensory MRTs and auditory WM. Indeed, they concluded that a faster speed of multisensory processing was unlikely to be constrained by children’s auditory WM abilities. This was partially supported by our results in the current study, as the relationship between auditory WM and multisensory MRTs is not as strong as the relationship between visual WM and multisensory MRTs, also suggesting that auditory processing in WM does not play an important role in multisensory motor speed tasks [2]. A further possible explanation of the bias for vision-based tasks in our present study could be associated with the maturation of audiovisual multisensory processing in the posterior superior temporal gyrus [83], as it has been suggested that children and adolescents with neurodevelopmental disorders such as dyslexia [84] and autism spectrum disorder (ASD) [83] demonstrate deficits. Indeed, the relative hypo-activation of the temporal gyrus has been associated with the incomplete maturation of the neural processes of audiovisual processing.

### 4.3. Contributions of Visual and Auditory Working Memory and Nonverbal IQ to MRT Multisensory Measures

The hypothesis regarding the contribution of combined cognitive measures of WM and nonverbal IQ tasks to MRTs of multisensory processing was supported by our Bayesian regression analyses that indicated visual STM and WM and nonverbal IQ consistently predicted MRTs as measured by AS, VA, AVS, and SLURP. Relationships between visual WM and fine motor skills have previously been explored in a longitudinal study conducted by Rigoli et al. who reported that visual WM predicts motor skill performance in primary school children [5]. Such findings are consistent with the results of the current study and past research in adults which found that visuospatial WM explained a significant portion of the variance in rates of MRT performance, whereas auditory WM did not significantly improve the model [2,85]. Additionally, past research has demonstrated that WM and intelligence are similar constructs [25,86], with success in completing WM tasks requiring individuals to hold, manipulate, and repeat information, and fluid intelligence tasks measuring nonverbal IQ also requiring reasoning in addition to storage and manipulation of information [86]. In line with this, our current findings suggest that both visual WM and nonverbal IQ require a better ability to manipulate information in WM, and that these abilities are heavily reliant on the motor speed of multisensory MRT tasks (i.e., faster motor speeds). Such findings highlight that visual WM and nonverbal IQ are the strongest predictors of multisensory MRTs, which is in line with the theoretical propositions that attentional control is an important aspect of the association between MRTs, WM, and intelligence [87,88,89]. A potential explanation of these theories was based on the idea that better attention control capacities lead to better WM and higher intelligence, which in turn result in shorter motor reaction times. This finding is also in line with the work of Voelke et al., who suggested that eye movements and shifts in attention mediate the relationships between MRT distributions and concepts of higher-order cognition such as WM capacity and reasoning [86].

## 5. Limitations

A notable strength of the current study relative to previous research investigating cognitive abilities and multisensory MRTs was the inclusion of both auditory and visual measures of WM, which provides a more complete picture of both domains. In addition, our study followed recent analytical recommendations [90,91,92] by using Bayesian probability statistics to assess the strength of the evidence of the alternative hypothesis. On the other hand, a major limitation of the current study was the decision not to time limit measures of WM (i.e., the focus was on capacity rather than response time), although there is previous evidence noting the importance of using time-limited tasks to assess working memory performance (see [2,93]). Thus, future studies should aim to include response time as well as capacity in the measurements of auditory and visual WM. Furthermore, visually presented digit span tasks in our study can be easily verbalized, and it is currently unknown whether the older students familiar with the symbols utilized additional verbal encoding to aid their performance. Thus, future research should aim to include visual WM using tasks with less verbalizable stimuli. In addition, we did not independently assess non-motor multisensory threshold detection times in the current study. Therefore, future studies may benefit from including both motor and non-motor multisensory threshold detection. Future research might also benefit from using other robust measures of motor reaction times, such as the GazePoint eye tracker to assess eye movements to nominated objects or sounds rather than motor reaction times to nonspecific Gaussian stimuli given that non-motor developmental literature in this area remains understudied.

## 6. Conclusions and Future Directions

To the best of our knowledge, the current study, which aimed to explore whether visual and auditory STM and WM and nonverbal intelligence contribute significantly to multisensory MRTs, was one of the first to investigate the effects of both domains of visual and auditory WM and nonverbal IQ on the rate of multisensory processing. Thus, our results are unique in providing preliminary insight into the importance of both visual and auditory WM, which contribute differentially to multisensory MRTs. The main findings from our Bayesian analyses revealed decisive evidence for grade differences in visual STM and WM, whereas auditory STM and WM showed no significant differences across the grades. Overall, performance on more complex visually-based tasks, such as the difficult items of the RCPM (i.e., those in Factor 3: Continuity and Reconstruction of Simple and Complex Structures) and digit span capacity on the visual WM task, improved across grade levels, apparently contributing significantly to faster MRTs for multisensory processing in elementary school children. Furthermore, our results confirm that visual rather than auditory processing is the most important cognitive driver associated with simple multisensory MRTs and that the enhanced development of visual WM is likely to contribute to the expected increase in cognitive abilities and multisensory motor processing seen with age. However, it is also important to note that, to date, few studies have reported the extent to which language abilities and vocabulary also contribute to age multisensory motor reaction times. Thus, future studies should examine factors such as receptive and expressive language skills that may contribute to the age-related development of multisensory MRTs.

## Figures and Tables

**Figure 1 brainsci-13-00270-f001:**
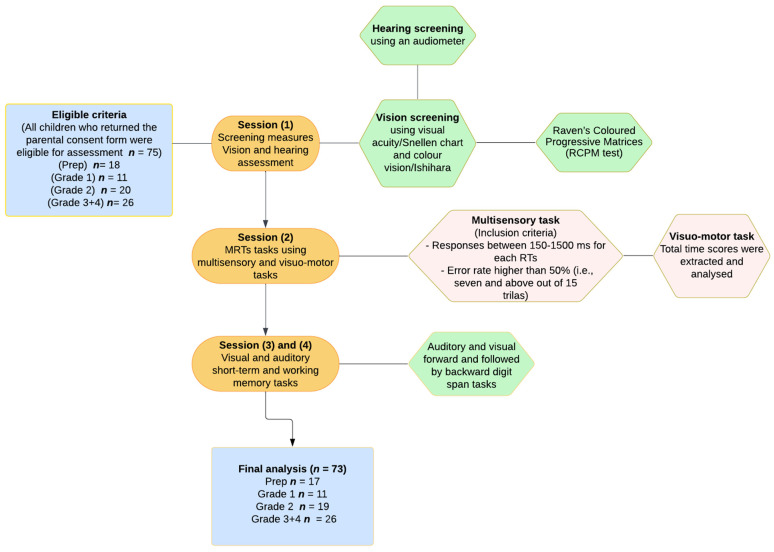
Flowchart illustrating eligibility criteria, children groups, and experimental series.

**Figure 2 brainsci-13-00270-f002:**
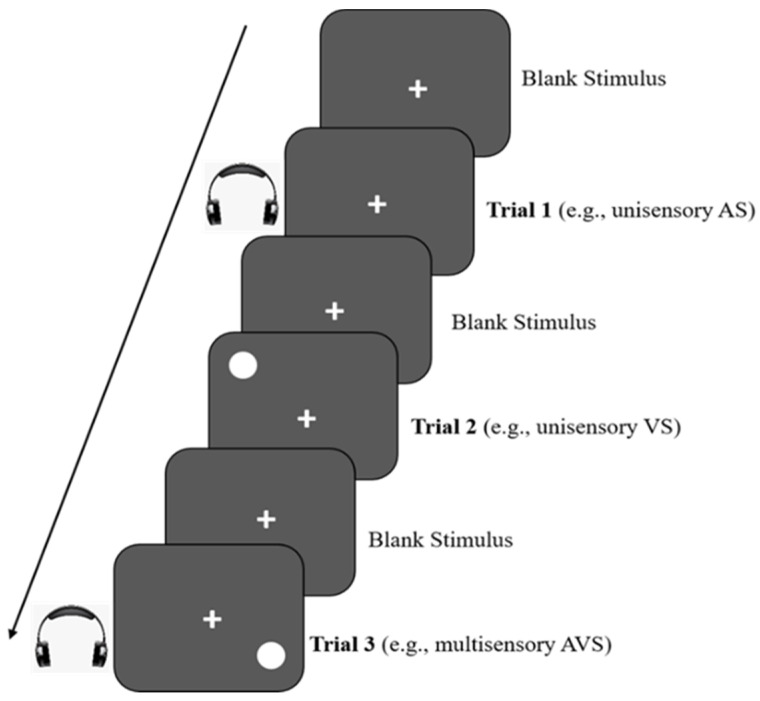
Example of the three types of stimuli used in Multisensory Trials [10].

**Figure 3 brainsci-13-00270-f003:**
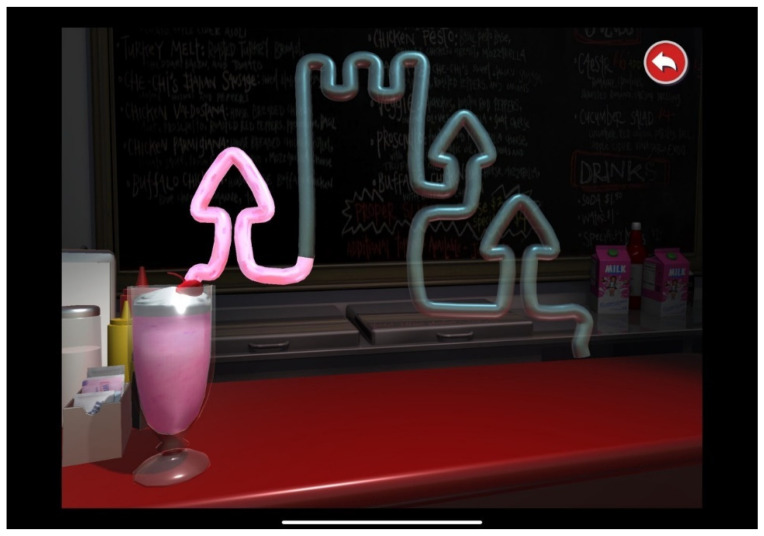
Example of castle shape of Lee-Ryan Hand Coordination Test (SLURP).

**Figure 4 brainsci-13-00270-f004:**
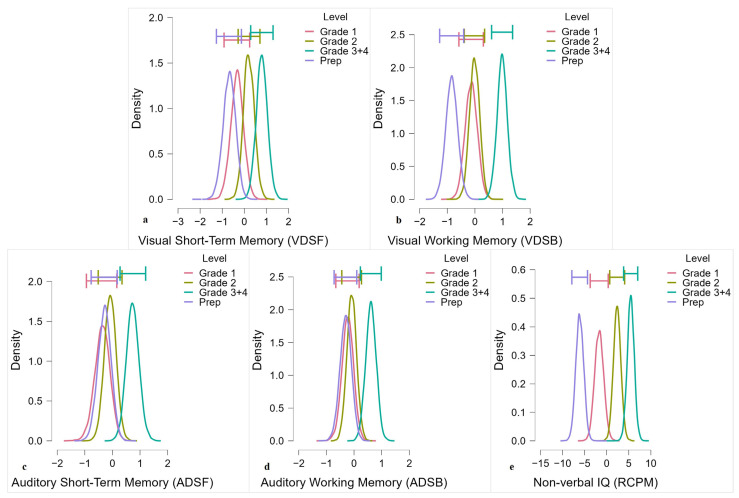
The model-averaged posterior distribution (horizontal bars show the 95% credible intervals around the median) for (**a**) visual digit span forward (**b**) visual digit span backward (**c**) auditory digit span forward (**d**) auditory digit span backward and (**e**) (RCPM) Raven’s Coloured Progressive Matrices.

**Table 1 brainsci-13-00270-t001:** Descriptive statistics of the mean age (±SD) and IQ raw score measures for each grade.

Grades N		AGE RANGE			Nonverbal IQ		
		Min.	Max.	M ± SD	Min.	Max.	M ± SD
Prep	17	5	6.71	5.71 ± 0.43	11	29	17.52 ± 5.58
Grade 1	11	6.4	7.47	6.85 ± 0.29	17	28	22.09 ± 3.64
Grade 2	19	7.56	8.77	8.01 ± 0.31	20	34	26.42 ± 3.83
Grade 3+4	26	8.58	10.9	9.84 ± 0.72	19	34	29.75 ± 3.47
Total	73						

Note. Non-verbal IQ assessed by the Raven’s Coloured Progressive Matrices, and scores range from 0–36.

**Table 2 brainsci-13-00270-t002:** Descriptive statistics for raw scores on digit span capacity for visual and auditory short-term and working memory tasks, and nonverbal IQ by grades.

					95% Credible Interval	
	Measure	Grade	M	SD	Lower	Upper
Visual and Auditory WM Tasks	Visual Short-Term Memory (VDSF)	Prep	3.857	1.657	2.9	4.814
		Grade 1	4.273	1.009	3.595	4.951
		Grade 2	4.95	0.887	4.535	5.365
		Grade 3+4	5.625	1.439	5.017	6.233
	Visual Working Memory (VDSB)	Prep	2.571	0.646	2.198	2.945
		Grade 1	3.364	0.505	3.025	3.703
		Grade 2	3.5	0.761	3.144	3.856
		Grade 3+4	4.583	1.283	4.042	5.125
	Auditory Short-Term Memory (ADSF)	Prep	4.929	1.141	4.27	5.587
		Grade 1	4.778	0.441	4.439	5.117
		Grade 2	5.2	1.056	4.706	5.694
		Grade 3+4	6.174	1.37	5.581	6.766
	Auditory Working Memory (ADSB)	Prep	3	0.577	2.651	3.349
		Grade 1	3.091	0.302	2.888	3.293
		Grade 2	3.3	0.801	2.925	3.675
		Grade 3+4	4.13	1.359	3.543	4.718
Non-verbal IQ	(RCPM)	Prep	17.529	5.580	14.660	20.399
		Grade 1	22.091	3.646	19.642	24.540
		Grade 2	26.421	3.834	24.573	28.269
		Grade 3+4	29.577	3.478	28.172	30.982

**Table 3 brainsci-13-00270-t003:** Bayesian post hoc comparisons for visual and auditory short-term and working memory, tasks, and nonverbal IQ by grade.

		Prior Odds	Posterior Odds	BF_10, U_	Error %
**a. VDSF**					
Grade 1	Grade 2	0.414	0.571	1.378	0.004
	Grade 3+4	0.414	2.381	5.748	7.92 × 10^−6^
	Prep	0.414	0.186	0.449	0.002
Grade 2	Grade 3+4	0.414	0.462	1.115	0.007
	Prep	0.414	1.342	3.241	0.009
Grade 3+4	Prep	0.414	9.567	23.096	1.61 × 10^−6^
**b. VDSB**					
Grade 1	Grade 2	0.414	0.162	0.391	0.003
	Grade 3+4	0.414	3.663	8.844	6.05 × 10^−6^
	Prep	0.414	5.52	13.326	7.78 × 10^−6^
Grade 2	Grade 3+4	0.414	7.698	18.585	9.73 × 10^−7^
	Prep	0.414	15.919	38.431	4.94 × 10^−7^
Grade 3+4	Prep	0.414	1561.507	3769.812	4.54 × 10^−9^
**c. ADSF**					
Grade 1	Grade 2	0.414	0.246	0.594	0.002
	Grade 3+4	0.414	3.129	7.555	1.35 × 10^−5^
	Prep	0.414	0.168	0.405	0.002
Grade 2	Grade 3+4	0.414	1.615	3.898	1.11 × 10^−6^
	Prep	0.414	0.168	0.405	0.003
Grade 3+4	Prep	0.414	2.632	6.355	3.43 × 10^−6^
**d. ADSB**					
Grade 1	Grade 2	0.414	0.188	0.454	0.003
	Grade 3+4	0.414	1.339	3.234	0.008
	Prep	0.414	0.168	0.406	0.002
Grade 2	Grade 3+4	0.414	1.146	2.766	0.009
	Prep	0.414	0.234	0.566	0.004
Grade 3+4	Prep	0.414	2.589	6.250	4.57 × 10^−6^
**e. Non-verbal IQ (RCPM)**					
Grade 1	Grade 2	0.414	3.459	8.352	8.81 × 10^−6^
	Grade 3+4	0.414	4638.691	11,198.791	1.19 × 10^−9^
	Prep	0.414	1.109	2.677	0.007
Grade 2	Grade 3+4	0.414	2.957	7.139	8.63 × 10^−7^
	Prep	0.414	2116.608	5109.943	1.96 × 10^−9^
Grade 3+4	Prep	0.414	3.83e+07	9.24e+07	1.66 × 10^−12^

Note. The posterior odds have been corrected for multiple comparisons by fixing to 0.5 the prior probability that the null hypothesis holds across all comparisons [60]. Individual comparisons are based on the default t-test with a Cauchy (0, r = 1/sqrt (2)) prior. The “U” in the Bayes factor denotes that it is uncorrected.

**Table 4 brainsci-13-00270-t004:** Bayesian Pearson Correlations for Total Sample.

Variable		Age	RCPM	AS	VS	AVS	SLURP	VDSF	VDSB	ADSF	ADSB
1. Age	Pearson’s r	—									
	BF₁₀	—									
2. RCPM	Pearson’s r	0.747 ***	—								
	BF₁₀	1.142 × 10^6^	—								
3. AS	Pearson’s r	−0.714 ***	−0.484 **	—							
	BF₁₀	159,221.512	31.340	—							
4. VS	Pearson’s r	−0.798 ***	−0.596 ***	0.844 ***	—						
	BF₁₀	4.764 × 10^7^	834.519	4.072 × 10^9^	—						
5. AVS	Pearson’s r	−0.785 ***	−0.556 ***	0.864 ***	0.883 ***	—					
	BF₁₀	1.623 × 10^7^	221.043	4.201 × 10^10^	5.906 × 10^11^	—					
6. SLURP	Pearson’s r	−0.670 ***	−0.498 **	0.497 **	0.434 *	0.516 **	—				
	BF₁₀	16,670.147	44.659	43.176	10.279	71.011	—				
7. VDSF	Pearson’s r	0.493 **	0.497 **	−0.409	−0.363	−0.322	−0.524 **	—			
	BF₁₀	39.077	43.677	6.384	2.859	1.557	87.322	—			
8. VDSB	Pearson’s r	0.779 ***	0.699 ***	−0.486 **	−0.561 ***	−0.537 ***	−0.587 ***	0.574 ***	—		
	BF₁₀	1.011 × 10^7^	67,790.264	33.189	257.850	128.710	619.187	389.869	—		
9. ADSF	Pearson’s r	0.545 ***	0.576 ***	−0.292	−0.474 *	−0.339	−0.428	0.575 ***	0.697 ***	—	
	BF₁₀	160.052	419.254	1.058	24.885	1.968	9.183	407.956	63,555.120	—	
10. ADSB	Pearson’s r	0.603 ***	0.616 ***	−0.411	−0.435 *	−0.392	−0.480 *	0.567 ***	0.724 ***	0.609 ***	—
	BF₁₀	1070.937	1710.254	6.535	10.454	4.658	28.584	313.522	269,457.545	1315.143	—

Note. Age = age in numbers; RCPM = nonverbal IQ of Raven; AS = MRTs of auditory stimuli; VS = MRTs of visual stimuli; AVS = MRTs of audiovisual stimuli; SLURP= visual motor skills; VDSF = visual digit span forward; VDSB = visual digit span backward; ADSF = auditory digit span forward; ADSB = auditory digit span backward. * BF₁₀ > 10, ** BF₁₀ > 30, *** BF₁₀ > 100.

**Table 5 brainsci-13-00270-t005:** Bayesian Pearson correlations (factors of nonverbal IQ, MRTs, and working memory tasks).

Variable		Age	AS	VS	AVS	SLURP	VDSF	VDSB	ADSF	ADSB
1. Factor1 (SPC)	Pearson’s r	0.325	−0.48 *	−0.379	−0.418	−0.182	0.241	0.124	0.128	0.118
	BF₁₀	1.619	28.49	3.695	7.49	0.367	0.603	0.258	0.263	0.251
2. Factor2 (DPC)	Pearson’s r	0.448 *	−0.231	−0.325	-0.213	-0.19	0.216	0.251	0.308	0.253
	BF₁₀	13.893	0.547	1.619	0.467	0.389	0.478	0.666	1.290	0.677
3. Factor3 (ContRecon)	Pearson’s r	0.681 ***	−0.445 *	−0.508 **	−0.469 *	−0.521 **	0.502 **	0.65 ***	0.525 **	0.500 **
	BF₁₀	28,678.991	13.098	57.718	22.031	80.786	49.631	6880.081	90.378	46.727
4. Factor4 (Reasoning)	Pearson’s r	0.610 ***	−0.364	−0.549 ***	−0.544 ***	−0.333	0.330	0.635 ***	0.631 ***	0.564 ***
	BF₁₀	1381.063	2.884	179.303	154.63	1.809	1.731	3629.421	3160.303	290.963

Note. Factor1 (SPC) = simple continuous pattern completion; Factor2 (DPC) = discrete pattern completion; Factor3 (ContRecon) = continuity and reconstruction of simple and complex structures; Factor4 (Reasoning) = reasoning by analogy; Age = age in numbers; AS = MRTs of auditory stimuli; VS = MRTs of visual stimuli; AVS = MRTs of audiovisual stimuli; SLURP= visual motor skills; VDSF = visual digit span forward; VDSB = visual digit span backward; ADSF = auditory digit span forward; ADSB = auditory digit span backward. * BF₁₀ > 10, ** BF₁₀ > 30, *** BF₁₀ > 100.

**Table 6 brainsci-13-00270-t006:** Bayesian multiple regressions for each multisensory MRT (auditory, visual, and audiovisual) and SLURP. Predictors were nonverbal IQ, auditory and visual short-term and working memory.

Model Predictors	P(M)	P(M|Data)	BF_M_	BF_10_	R^2^
**a. Auditory RT**					
VDSF + VDSB	0.031	0.138	4.952	1.000	0.304
VDSF + RCPM	0.031	0.103	3.544	0.745	0.296
VDSF	0.031	0.078	2.641	0.570	0.251
VDSF + VDSB + RCPM	0.031	0.074	2.493	0.540	0.321
VDSF + ADSB	0.031	0.073	2.455	0.533	0.288
VDSF + ADSB + RCPM	0.031	0.057	1.857	0.410	0.314
VDSF + VDSB + ADSB	0.031	0.050	1.645	0.366	0.311
VDSB	0.031	0.046	1.498	0.335	0.237
VDSF + VDSB + ADSF	0.031	0.042	1.364	0.306	0.307
VDSB + RCPM	0.031	0.034	1.106	0.250	0.267
**b. Visual RT**					
VDSB + RCPM	0.031	0.296	13.011	1.000	0.436
VDSB + ADSF + RCPM	0.031	0.113	3.957	0.383	0.448
VDSB + ADSB + RCPM	0.031	0.088	3.000	0.298	0.442
VDSF + VDSB + RCPM	0.031	0.088	2.996	0.298	0.442
ADSF + RCPM	0.031	0.059	1.947	0.200	0.403
ADSB + RCPM	0.031	0.048	1.568	0.163	0.399
ADSF + ADSB + RCPM	0.031	0.035	1.139	0.120	0.423
VDSB + ADSF + ADSB + RCPM	0.031	0.034	1.086	0.114	0.451
VDSF + VDSB + ADSF + RCPM	0.031	0.033	1.072	0.113	0.450
VDSF + RCPM	0.031	0.030	0.943	0.100	0.388
**c. Audiovisual RT**					
VDSB + RCPM	0.031	0.260	10.914	1.000	0.380
RCPM	0.031	0.142	5.118	0.544	0.329
ADSB + RCPM	0.031	0.100	3.450	0.385	0.358
VDSB + ADSB + RCPM	0.031	0.072	2.392	0.275	0.383
VDSF + RCPM	0.031	0.068	2.264	0.261	0.349
VDSF + VDSB + RCPM	0.031	0.068	2.263	0.261	0.382
VDSB + ADSF + RCPM	0.031	0.064	2.115	0.245	0.380
ADSF + RCPM	0.031	0.049	1.587	0.187	0.341
VDSF + ADSB + RCPM	0.031	0.032	1.021	0.122	0.364
ADSF + ADSB + RCPM	0.031	0.027	0.858	0.103	0.360
**d. SLURP**					
VDSF + VDSB	0.031	0.142	5.137	1.000	0.390
VDSB + RCPM	0.031	0.132	4.720	0.930	0.388
VDSB	0.031	0.115	4.044	0.812	0.342
VDSF + VDSB + RCPM	0.031	0.087	2.940	0.609	0.413
VDSF + RCPM	0.031	0.053	1.725	0.371	0.362
VDSF + VDSB + ADSB	0.031	0.039	1.252	0.273	0.391
VDSF + VDSB + ADSF	0.031	0.038	1.228	0.268	0.390
VDSB + ADSB + RCPM	0.031	0.038	1.210	0.264	0.390
VDSB + ADSF + RCPM	0.031	0.035	1.140	0.250	0.388
VDSB + ADSB	0.031	0.034	1.081	0.237	0.349

Note. SLURP= visual motor skills; RCPM = Raven’s Coloured Progressive Matrices; VDSF = visual digit span forward; VDSB = visual digit span backward; ADSF = auditory digit span forward; ADSB = auditory digit span backward.

**Table 7 brainsci-13-00270-t007:** Posterior summaries of regression coefficients.

Coefficient	P(incl)	P(incl|data)	BF_inclusion_	Mean	SD	95% Credible Interval	
						Lower	Upper
**a. Auditory RT**							
Intercept	1.000	1.000	1.000	872.411	15.869	839.149	903.814
VDSF	0.500	0.796	3.897	−25.81	18.169	−55.13	0.000
VDSB	0.500	0.552	1.233	−17.648	21.373	−61.011	1.293
ADSF	0.500	0.242	0.319	0.827	8.372	−18.649	24.115
ADSB	0.500	0.375	0.599	−8.698	16.764	−52.398	8.372
RCPM	0.500	0.451	0.823	−2.000	3.027	−8.933	0.627
**b. Visual RT**							
Intercept	1.000	1.000	1.000	905.492	13.232	880.261	932.099
VDSF	0.500	0.268	0.366	−2.786	7.790	−24.684	7.327
VDSB	0.500	0.743	2.890	−26.288	20.480	−60.510	0.826
ADSF	0.500	0.347	0.532	−6.067	11.848	−40.842	1.069
ADSB	0.500	0.296	0.420	−5.115	12.490	−46.959	1.706
RCPM	0.500	0.944	16.95	−7.623	3.230	−12.585	0.000
**c. Audiovisual RT**							
Intercept	1.000	1.000	1.000	821.221	13.209	794.332	846.998
VDSF	0.500	0.251	0.335	−1.893	6.667	−23.855	6.914
VDSB	0.500	0.551	1.230	−14.275	16.775	−49.499	0.304
ADSF	0.500	0.221	0.284	−0.709	6.666	−23.327	10.434
ADSB	0.500	0.296	0.420	−4.360	11.174	−33.149	8.531
RCPM	0.500	0.980	47.980	−8.881	2.996	−15.301	−3.594
**d. SLURP**							
Intercept	1.000	1.000	1.000	66.385	2.142	62.335	70.476
VDSF	0.500	0.510	1.042	−1.521	1.983	−5.794	0.400
VDSB	0.500	0.789	3.745	−4.553	3.226	−9.835	0.000
ADSF	0.500	0.235	0.307	−0.100	1.245	−3.044	3.128
ADSB	0.500	0.259	0.350	−0.403	1.621	−4.913	2.314
RCPM	0.500	0.541	1.180	−0.460	0.565	−1.747	0.000

Note. SLURP = visual motor skills; RCPM = Raven’s Coloured Progressive Matrices; VDSF = visual digit span forward; VDSB = visual digit span backward; ADSF = auditory digit span forward; ADSB = auditory digit span backward.

## Data Availability

All data are available upon request.

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
