# Peer review of "The Contribution of Visual and Auditory Working Memory and Non-Verbal IQ to Motor Multisensory Processing in Elementary School Children"

_brainsci, 2023, doi:10.3390/brainsci13020270_

Round 1

Reviewer 1 Report

Comments and Suggestions for Authors

The article is devoted to the study of the contribution of some factors to motor multisensory processing in elementary school children. The authors chose visual and auditory working memory and non-verbal IQ as the factors. 

In the Introduction, the authors substantiate in detail the relevance of the chosen topic for the study, outline the general and specific objectives of the study.

In the Materials and Methods, the authors clearly describe the design of the experiment, all of its methodology. 

The Results are presented in detail.

In the Discussion, the authors compare their results with results already available in research papers.

Importantly, the authors attach supplementary material to the article, which makes it easier to read.

My only recommendation is to add specific details in the Conclusions section.

Author Response

To the Assigned Editor,

Ms. Della Diao

Thank you for giving us the opportunity to submit a revised draft of our manuscript for publication in the Journal of Brain Sciences/ Sensory and Motor Neuroscience section. We appreciate the time and effort that you and the other reviewers have devoted to reviewing our manuscript and making insightful comments. Please see below, where we have responded to each comment for the three reviewers. Please see the attachment.

We have also highlighted the associated changes within the manuscript.

 Point 1: The article is devoted to the study of the contribution of some factors to motor multisensory processing in elementary school children. The authors chose visual and auditory working memory and non-verbal IQ as the factors. In the Introduction, the authors substantiate in detail the relevance of the chosen topic for the study, outline the general and specific objectives of the study. In the Materials and Methods, the authors clearly describe the design of the experiment, all of its methodology. The Results are presented in detail.In the Discussion, the authors compare their results with results already available in research papers. Importantly, the authors attach supplementary material to the article, which makes it easier to read.

My only recommendation is to add specific details in the Conclusions section.

Response 1: 

We thank the Reviewer for appreciating our manuscript and following his/her advice, we have expanded the conclusion to read:

“To the best of our knowledge, the current study which aimed to explore whether visual and auditory STM and WM and nonverbal intelligence contribute significantly to multisensory MRT, was one of the first to investigate the effects of both domain visual and auditory WM and nonverbal IQ on rate of multisensory processing. Thus, our results are unique in providing preliminary insight into the importance of both visual and auditory WM, which contribute differentially to multisensory MRT. The main findings from our Bayesian analyses revealed decisive evidence for grade differences in visual STM and WM, while auditory STM and WM showed no significant differences across grades. Overall, performance on more complex visually based tasks such as the difficult items of the RCPM (i.e., those in Factor 3: Continuity and Reconstruction of Simple and Complex Structures), and digit span capacity in the visual WM task improved across grade levels, apparently contributing significantly to faster MRT for multisensory processing in elementary school children.” (please see pg. 20).

Reviewer 2 Report

Comments and Suggestions for Authors

The paper “The Contribution of Visual and Auditory Working Memory and Non-Verbal IQ to Motor Multisensory Processing in Early Elementary School Children” investigates the relationship between age-related cognitive development and multisensory-motor processing. To this aim the authors analyzed data from 75 5-10 aged children and concerning working memory, non-verbal reasoning, and visuomotor and multisensory tasks. The results showed that the performance in visual memory tasks, but not in auditory tasks, was a predictor of multisensory motor response times. The study sounds timely and worth. The results are straightforward. The manuscript offers a general overview of the findings: The discussion could be strengthened by further specifying the study's potential impact in a broader framework. In particular, the brain-related paragraph of the discussion (4.1.) correctly highlights the age-related relationship between brain maturation and the difference between auditory and visual memory. The authors seem to suggest that the processing of visual input starts in earlier phases of development.  To support claim, some previous studies are cited, but a deep/full overview of the anatomo-functional relationship between neural maturation of the (brain's) visual system and cognitive/behavioral skills (e.g. Ionta 2021, Frontiers in Human Neuroscience) seems somehow missing. Providing a more mechanistic interpretation of how the maturation of the visual system could explain the advantage of visual memory tasks in the present study could be beneficial to better understand how the present study sits in a broader framework. In addition, it might be worth noting that the temporal gyrus plays a major role in audio-visual multisensory tasks, such as reading, including deficits associated with temporal hypoactivations in children, adolescents, young adults, and adults (Farah et al 2021, Frontiers in Psychology). On this basis, it could be proposed that the bias for vision-based tasks in the present study could be associated with an incomplete maturation of a audio-visual integration area such as the temporal gyrus. What is the authors' account?

Author Response

To the Assigned Editor, and the reviewer,

Thank you for giving us the opportunity to submit a revised draft of our manuscript for publication in the Journal of Brain Sciences/ Sensory and Motor Neuroscience section. We appreciate the time and effort that you and the other reviewers have devoted to reviewing our manuscript and making insightful comments that we have now incorporated and believe has greatly improved our manuscript. Please see below, where we have responded to each comment for the three reviewers. Please see also the attachment.  We have also highlighted the associated changes within the manuscript.

Point 1: 

The paper “The Contribution of Visual and Auditory Working Memory and Non-Verbal IQ to Motor Multisensory Processing in Early Elementary School Children” investigates the relationship between age-related cognitive development and multisensory-motor processing. To this aim the authors analyzed data from 75 5-10 aged children and concerning working memory, non-verbal reasoning, and visuomotor and multisensory tasks. The results showed that the performance in visual memory tasks, but not in auditory tasks, was a predictor of multisensory motor response times. The study sounds timely and worth. The results are straightforward.

(1) The manuscript offers a general overview of the findings: The discussion could be strengthened by further specifying the study's potential impact in a broader framework. In particular, the brain-related paragraph of the discussion (4.1.) correctly highlights the age-related relationship between brain maturation and the difference between auditory and visual memory. The authors seem to suggest that the processing of visual input starts in earlier phases of development.  To support claim, some previous studies are cited, but a deep/full overview of the anatomo-functional relationship between neural maturation of the (brain's) visual system and cognitive/behavioral skills (e.g. Ionta 2021, Frontiers in Human Neuroscience) seems somehow missing. Providing a more mechanistic interpretation of how the maturation of the visual system could explain the advantage of visual memory tasks in the present study could be beneficial to better understand how the present study sits in a broader framework.

Response 1: 

We thank the Reviewer for appreciating our manuscript and we would like to thank him/her also for this excellent recommending.

  • We have added further interruption to the dissuasion (4.1) regarding the study's potential impact, as suggested to read

“Furthermore, more recent cognitive neuroscience models for visual perception indicate that while there are multiple interconnections between the two major functional visual streams (i.e., dorsal and ventral streams [69,70], different visuomotor sub-pathways also exist within the longitudinal fasciculi of the dorsal stream[71], with the ventro-dorsal stream playing a role in the online control of action, and the dorso-dorsal stream being involved in higher-level cognitive processes such as action understanding [69,72]. Importantly, the anatomo-functional findings of Ionta study also demonstrate that the neural maturation of the visual system results in improvement in a variety of visual skills such as visual exploration, visual field awareness and motion sensitivity as well as cognitive abilities such as attention, working memory, and visuomotor eye-hand coordination [69]. Furthermore, age-related increases in activity in the frontal areas during visual WM tasks and increasing task demand have been demonstrated in children [33,67,73].” (pg. 17). On this basis, we have included four additional references as following

Ionta, S. (2021). Visual Neuropsychology in Development: Anatomo-Functional Brain Mechanisms of Action/Perception Binding in Health and Disease [Review]. Frontiers in Human Neuroscience, 15. https://doi.org/10.3389/fnhum.2021.689912

 Kravitz, D. J., Saleem, K. S., Baker, C. I., Ungerleider, L. G., & Mishkin, M. (2013). The ventral visual pathway: an expanded neural framework for the processing of object quality. Trends in Cognitive Sciences, 17(1), 26-49. https://doi.org/https://doi.org/10.1016/j.tics.2012.10.011

Thiebaut de Schotten, M., Dell’Acqua, F., Forkel, S., Simmons, A., Vergani, F., Murphy, D. G. M., & Catani, M. (2011). A Lateralized Brain Network for Visuo-Spatial Attention. Nature Precedings. https://doi.org/10.1038/npre.2011.5549.1

Rizzolatti, G., & Matelli, M. (2003). Two different streams form the dorsal visual system: anatomy and functions. Experimental Brain Research, 153(2), 146-157. https://doi.org/10.1007/s00221-003-1588-0

Point 2: ) In addition, it might be worth noting that the temporal gyrus plays a major role in audio-visual multisensory tasks, such as reading, including deficits associated with temporal hypoactivations in children, adolescents, young adults, and adults (Farah et al 2021, Frontiers in Psychology). On this basis, it could be proposed that the bias for vision-based tasks in the present study could be associated with an incomplete maturation of a audio-visual integration area such as the temporal gyrus. What is the authors' account?

Response 2: 

  • We also would like to thank you for pointing this out. We have now added more details to the discussion (4.2) to read

“A further possible explanation of the bias for vision-based tasks in our present study could be associated with the maturation of audiovisual multisensory processing in the posterior superior temporal gyrus [83] where it has been suggested that children and adolescents with neurodevelopmental disorders such as dyslexia [84] and autism spectrum disorder (ASD) [83] demonstrate deficits. Indeed, relative hypo-activation of the-temporal gyrus has been associated with incomplete maturation of neural processes of audiovisual processing.” (pg.18). We have also included two more references 

Cuppini, C., Ursino, M., Magosso, E., Ross, L. A., Foxe, J. J., & Molholm, S. (2017). A Computational Analysis of Neural Mechanisms Underlying the Maturation of Multisensory Speech Integration in Neurotypical Children and Those on the Autism Spectrum [Original Research]. Frontiers in Human Neuroscience, 11. https://doi.org/10.3389/fnhum.2017.00518

Farah, R., Ionta, S., & Horowitz-Kraus, T. (2021). Neuro-Behavioral Correlates of Executive Dysfunctions in Dyslexia Over Development From Childhood to Adulthood [Review]. Frontiers in Psychology, 12. https://doi.org/10.3389/fpsyg.2021.708863

Reviewer 3 Report

Comments and Suggestions for Authors

This is an interesting piece of research that presents some unique findings. I found the introduction quite clear and understood the purpose and aims of the research. There were a coupld of queries about the methods & results though (see below) as well as some of the ways in which the article was written. Nevertheless, I think this research has value and contributes to our knowledge in this area.

Amendments:

- There seem to be some changes in participant numbers mentioned. Between recruitment and final analysis. I assume that these were children who dropped out for some reason, but it’s not clear why they dropped out, or when in the process. This should be mentioned in the procedure.

 - The results are extensive and complicated to those not familiar with this type of analysis. I would suggest considering reducing this section to display only the main findings, or at least having a summary at the end of the results section to provide an overview of the findings.

 - I would usually report numbers to 2 significant digits, not three.

 - P18 Final para section beginning “i.e.,…” should be inside brackets

- P19. Line 1 There is no concluding round bracket.

There are some odd phrasings throughout this article, that appear to be due to the adaptation from text referencing to numerical referencing. To me it is bad use of English to supplement numbers directly for names, and sometimes sentences need to be re-written. Therefore, I would suggest that the following phrases are amended.

P2. Para 2. Sentence 3 states “…with (25) previously suggesting….”. Sentence 4 cannot begin with a number “(26) also found…”. The final sentence states “leading (29) and (16) to conclude…”

P2. Para 3.  The first sentence including “…a study that was conducted by (30)…”. The second sentence it refers to ‘The authors…’ when none have been named. The third sentence “(3,12,31) have also shown…”

P2. Para 4. Sentence 2 “For example, (32) used…” Sentence 3 “The authors…” is used when none have been directly referred to.

P7 Section 2.5 Para 2. “…according to (52) as evidence…”

P9. Para 2. Final sentence “…as associated with (28) factors…)

P12. Para 1.Final sentence “…according to (28)...”.

P17. Para 1. Line 2 “In line witth this, (33) reported…” Final Para. Final Sentence “For example, (3, 12) also…”

P18. Para 2. Line 2 “…conducted by (5) who reported…” Final line “…the work of (78) who suggested…”

Author Response

To the Assigned Editor, and reviewer, 

Thank you for giving us the opportunity to submit a revised draft of our manuscript for publication in the Journal of Brain Sciences/ Sensory and Motor Neuroscience section. We appreciate the time and effort that you and the other reviewers have devoted to reviewing our manuscript and making insightful comments. Please see below, where we have responded to each comment. Please see also the attachment.

We have also highlighted the associated changes within the manuscript.

Point 1: This is an interesting piece of research that presents some unique findings. I found the introduction quite clear and understood the purpose and aims of the research. There were a couple of queries about the methods & results though (see below) as well as some of the ways in which the article was written. Nevertheless, I think this research has value and contributes to our knowledge in this area.

Amendments:

1 - There seem to be some changes in participant numbers mentioned. Between recruitment and final analysis. I assume that these were children who dropped out for some reason, but it’s not clear why they dropped out, or when in the process. This should be mentioned in the procedure.

Response 1:  We thank the reviewer for the valuable comments and thank you for raising this point regarding the need to clarify the final sample size. We have now clarified this to read

“The data of 2 children (one in the Prep year, and one in Grade 1 group) whose error score was greater than 50% in either the AS or VS trials were excluded (see Figure 1 for details)’ (please see pg.7 Line 25).

Point 2: The results are extensive and complicated to those not familiar with this type of analysis. I would suggest considering reducing this section to display only the main findings, or at least having a summary at the end of the results section to provide an overview of the findings.

Response 2: We thank the reviewer for revising our results and we have now provided a summary of the results at the end of the results section as suggested, to read

“To summarize, our results comparing grade differences show that performance on visual based tasks such as Visual Digit Span Forward (VDSF) and Backward (VDSB) tasks and nonverbal IQ (RCPM) are significantly different supporting the alternative hypothesis. However, Auditory Digit Span Forward (ADSF) and Backward (ADSB) tasks showed no evidence of differences. In addition, Bayesian correlation showed decisive evidence of age-related correlations between the RCPM test scores and item difficulty, visual WM and multisensory MRT measures, but no significant correlations with auditory WM and Multisensory MRTs. Finally, Bayesian regression demonstrated that visual STM and WM together with nonverbal IQ consistently predicted multisensory MRT for AS, VS, AVS and time to complete the SLURP visuo-motor processing task” (please see pg. 14 Line 42).

Point 3: I would usually report numbers to 2 significant digits, not three.

Response 3:  We thank the reviewer for this comment. We acknowledge that it is a general convention of frequentist statistics to report significance to 2 decimals when reporting statistical outcomes. However, unlike p values in frequentist statistics, the Bayes factor in Bayesian statistics provides an indication of the strength of the evidence (effect size) for the alternative (experimental) hypothesis against rival (prior) models meaning that there is value in reporting to the third decimal, which is in line with the Bayesian Analysis Reporting Guidelines and JASP Bayesian reporting guidelines [53,54] to ensure transparency and allow for critical appraisal of our hypothesis. On this basis, we have included two additional references (Kruschke, 2021; van Doorn et al., 2021), and this has now been added and clarified in the text (please see pg. 7 and 8).

Point 4: P18 Final para section beginning “i.e.,…” should be inside brackets

Response 4: Thank you for noting this mistake. We have now amended all typos we identify on both pg.3 and pg.19.  

Point 5:  P19. Line 1 There is no concluding round bracket.

Response 5: Thank you for pointing this out. We have now checked and there are no more references in the conclusion

Point 6: 

There are some odd phrasings throughout this article, that appear to be due to the adaptation from text referencing to numerical referencing. To me it is bad use of English to supplement numbers directly for names, and sometimes sentences need to be re-written. Therefore, I would suggest that the following phrases are amended.

P2. Para 2. Sentence 3 states “…with (25) previously suggesting….”. Sentence 4 cannot begin with a number “(26) also found…”. The final sentence states “leading (29) and (16) to conclude…”

P2. Para 3.  The first sentence including “…a study that was conducted by (30)…”. The second sentence it refers to ‘The authors…’ when none have been named. The third sentence “(3,12,31) have also shown…”

P2. Para 4. Sentence 2 “For example, (32) used…” Sentence 3 “The authors…” is used when none have been directly referred to.

P7 Section 2.5 Para 2. “…according to (52) as evidence…”

P9. Para 2. Final sentence “…as associated with (28) factors…)

P12. Para 1.Final sentence “…according to (28)...”.

P17. Para 1. Line 2 “In line witth this, (33) reported…” Final Para. Final Sentence “For example, (3, 12) also…”

P18. Para 2. Line 2 “…conducted by (5) who reported…” Final line “…the work of (78) who suggested…”

Response 6: We appreciate the reviewer for noting these inconsistencies, we have now modified all sentences as suggested on (pg. 2, 3, 7, 9, 12, 17, 18 and lastly pg19), and we have also highlighted them in the manuscript.

Round 2

Reviewer 2 Report

Comments and Suggestions for Authors

accept